# Understanding Low-Speed Streaks and Their Function and Control through Movable Shark Scales Acting as a Passive Separation Control Mechanism

**DOI:** 10.3390/biomimetics9070378

**Published:** 2024-06-22

**Authors:** Leonardo M. Santos, Amy Lang, Redha Wahidi, Andrew Bonacci, Sashank Gautam

**Affiliations:** Department of Aerospace Engineering, The University of Alabama, Tuscaloosa, AL 35487, USA

**Keywords:** shark skin, passive flow control, streak spanwise spacing, low-speed streaks, turbulence

## Abstract

The passive bristling mechanism of the scales on the shortfin mako shark (*Isurus oxyrinchus*) is hypothesized to play a crucial role in controlling flow separation. In the hypothesized mechanism, the scales are triggered in response to patches of reversed flow at the onset of separation occurring in the low-speed streaks that form in a turbulent boundary layer. The two goals of this investigation were as follows: (1) to measure the reversing flow occurring within the low-speed streaks in a separating turbulent boundary layer; (2) to understand the passive flow control mechanism of movable shark skin scales that inhibit reversing flow within the low-speed streaks. Experiments were conducted using digital particle image velocimetry (DPIV). DPIV was used to analyze the flow in a turbulent boundary layer subjected to an adverse pressure gradient formation over both a smooth flat plate and a flat plate on which shark skin specimens were affixed. The experimental analysis of the flow over the smooth flat plate corroborated the findings of previous direct numerical simulation studies, which indicated that the average spanwise spacing of the low-speed streaks increases in the presence of adverse pressure gradients upstream of the point of separation. However, the characteristics of the flow over the shark skin specimen more closely resemble that of a zero-pressure gradient turbulent boundary layer. A comparative analysis of the width and velocity of the reversed streaks between flat plate and shark skin cases reveals that the mean spanwise spacing decreases, and thus, the number of streaks increases over the shark skin. Additionally, the reversed streaks observed over shark scales are thinner and the highest negative velocity within the streaks falls within the range required to bristle the scales.

## 1. Introduction

Research has consistently shown that the passive bristling of movable scales on the shortfin mako shark (*Isurus oxyrinchus*) effectively controls flow separation [1,2,3,4,5,6,7,8,9,10]. Flow separation typically occurs when the reversing flow overcomes the oncoming low-momentum flow near a surface. At the onset of separation in a turbulent boundary layer and before a global separation occurs, the reversed flow first occurs in the regions of low-speed streaks due to these being the regions of lowest momentum in the vicinity of the surface [11]. We hypothesize that these patches of reversing flow cause the scales to bristle, thereby impeding the reversed flow from further moving upstream. The bristled scales maintain a higher momentum flow near the wall, enabling the fluid to overcome the adverse pressure gradient and maintain its forward motion, thereby controlling flow separation.

The formation of horseshoe vortices in a turbulent boundary layer is one of the earliest proposed models for fundamental flow structures occurring in turbulent flow near walls. According to this model, small perturbations cause the flow between the vortex loop and legs to be pushed upstream, causing spanwise vortex lines close to the wall to stretch upward (Figure 1) [12]. These vortex lines travel faster than the legs (streamwise vortices), which are still in contact with the surface [13,14]. Confirmation of the streamwise streaks of low-speed momentum between vortex legs was obtained by injecting dye near the wall in turbulent flow [15], while the streamwise vortices exist between parallel low and high-speed streaks distributed in the spanwise direction. A more detailed investigation using hydrogen bubbles revealed that long streaks oscillate in both spanwise and streamwise directions, and are sporadically ejected into the outer layer. This periodic process, known as “bursting”, entails a sequence of lifting, stretching, and oscillation due to instability, followed by the breakdown and ejection of low-speed streaks (LSSs) into the outer layer when turbulence production reaches its maximum [16]. Notably, these low-speed streaks oscillate faster and become shorter when the boundary layer is subjected to an adverse pressure gradient (APG). Furthermore, the long streamwise streaks characterized by low momentum are also observed in the buffer layer, with a mean spanwise spacing ($\bar{\lambda^{+}}$) of approximately 100 ± 20 viscous scale units $\left(\bar{\lambda^{+}}=\frac{\lambda \;u_{\tau }}{\nu }\right)$, (λ is the spanwise distance between low-speed streaks, $u_{\tau }$ is the friction velocity, and ν is the kinetic viscosity). However, these streaks become less visible with increasing distance from the wall [15].

Blackwelder and Eckelmann’s hypothesis [17] posits that low-speed streaks form as fluid is pumped away from the wall by counter-rotating streamwise vortices parallel to the wall, with cores at a distance of approximately y^+^ = 20 away from the wall $\left(y^{+}=\frac{y\;u_{\tau }}{\nu }\right)$, where y is the distance from the wall, $u_{\tau }$ is the friction velocity, and ν is the kinematic viscosity. Experimental flow visualization studies show counter-rotating streamwise vortices with a small slope traveling downstream and attached to a transverse vortex, validating both the horseshoe vortex model [18] and Blackwelder’s hypothesis [17]. Furthermore, a study using a laser sheet at an inclined plane of 45° to the flow direction revealed inclined structures with a stronger vorticity vector in the streamwise direction but no vortex pairs. Conversely, a plane inclined at 135° to the flow direction showed vortex pairs of various diameters moving towards the plate as the hairpin vortex passed the laser-lighted inclined plane [19]. Both planes concurred with the hairpin vortex model as a dominating coherent structure within the turbulent boundary layer, and as the Reynolds numbers based on the characteristic length (Re_x_) increased, sparser hairpins occurred [19]. For y^+^ < 30, the streaks merge and split to maintain a nearly constant spanwise spacing, while a residual of the low-speed fluid remains close to the wall after a streak burst. Moreover, a flow visualization study using hydrogen bubbles revealed an increase in total turbulent kinetic energy [20]. Space–time correlation analyses between bursts indicated independent persistence near the wall regardless of Re_x_. Average streak persistence was approximately 480 viscous time scales $\left(t_{p}^{+}=\frac{t{\;u}_{\tau }^{2}\;}{{\;U}_{0}^{2}}\right)$, where t is the time between bursts and U_0_ is the free stream velocity [21,22]. The residual low-speed fluid was found to either merge with an old streak or initiate a new one. Beyond y^+^ > 30, spanwise spacing gradually increases with distance from the wall as vortical structures become more complex [21].

Complex vortical structure relationships have been described by a different conceptual model for hairpin-shaped vortices for y^+^ < 100. Following the burst of the streak, the formation of an unstable shear layer along the top and sides of the residual streak initiates the development of counter-rotating vortices. These vortices expel fluid from the vicinity of the wall, consequently giving rise to low-speed streaks between their legs. Preservation of the low-speed streaks after bursting is attributed to the occurrence of hairpins in packets, which explains the agglomeration of vortex loop heads in the outer region that represent the bulges between the outer layer of the boundary layer and the free stream flow [22]. Another explanation for streak formation was offered in a high shear rate direct numerical simulation (DNS) study that shows streaks near a shear layer, suggesting that a high shear rate might trigger turbulent flow to generate coherent structures similar to those in the viscous sublayer, especially low-speed streaks [23,24]. Streaks were also observed near the free-slip surface under a high shear rate, indicating that a solid boundary is not mandatory for generating streaks. Researchers have also attempted to link the formation of streaky patterns under rough wall conditions with a high shear rate, hypothesizing that high shear could generate self-organized turbulent structures moving away from the wall. Additionally, streaks have also been observed in the logarithmic region, with DPIV measurements, revealing a thickening and more uniform low momentum region near the top of the logarithmic region. The legs of hairpin vortices became more circular and further apart, leading to the hypothesis that the hairpin structure was the most common coherent structure to form in the logarithmic region [25,26,27].

The impact of surface roughness on fluid flow must be considered by establishing a scaling that captures how roughness characteristics influence the flow when the height of the surface irregularities exceeds the thickness of the viscous sublayer. The spanwise spacing of low-speed streaks is directly proportional to the height of the roughness element ($k_{s}$). As the height of the element increases, so does the spanwise spacing. The spanwise spacing of low-speed streaks depends on local effective eddy viscosity for rough walls. Research has shown it to be equivalent to four times the height of the roughness element near the wall ($\lambda =4k_{s}$) or four times the distance from the wall to the buffer region (λ=4 ∗ y), compared to $\bar{\lambda^{+}}$ ≈ 100 ± 20 for smooth walls [28,29].

The response of a boundary layer to a strong APG was examined in great detail using a DNS simulation, both with and without flow separation being induced [30]. Vortical structures weaken as they move downstream in the presence of a strong APG without separation due to the APG’s damping effect. Also, spanwise spacing $\left(\lambda =\frac{100\;\nu }{u_{\tau }}\right)$, where 100 is the average spanwise spacing ($\bar{\lambda^{+}}$) in a zero-pressure gradient flow, increased by as much as 30%, depending on the local friction velocity ($u_{\tau }$). In the presence of an APG inducing flow separation, the peak reversing flow velocity of the low-speed streaks was found to be around 20–30% of the freestream flow, and the spanwise spacing increases before vanishing at the beginning of the separation line. Finally, streaks were again visible downstream of flow reattachment [30]. There was also an indication that streaks would begin forming around the reattachment point if the backflow was sufficiently strong [30]. It is hypothesized that the LSS width must be comparable to the scale crown width to initiate the scale bristling process when the skin is properly sized dimensionally to the flow being controlled, such as on the body of a mako at burst swimming speeds. These patches of reversing flow are postulated to be the passive mechanism that actuates the scales to impede flow reversal, thereby eliminating, reducing, or delaying global flow separation.

Based on confirmation by previous studies that shortfin mako shark scales from the flank region mounted on a smooth flat plate or hydrofoil can control flow separation [3,4,10], we hypothesize that small localized bristling of the scales induced in the LSS regions impede the development of more global reversing flow and prevent large-scale separation that leads to high-pressure drag. Biological measurements of the scales covering the shortfin mako showed that the region behind the gills (flank region, Figure 2) has the highest bristling capability. Biologists found that the scale here could be manually bristled up to 50° and the rest at 45° upon release [31]. The flank region scales have a neck that connects to a small triangular base embedded in dermal tissue, resulting in flexibility within the skin- These scales have a long, thin crown with three riblets and a crown width of approximately 170 µm (Figure 3).

An experimental study of the flow over a shortfin mako shark skin specimen demonstrated that the scales bristle due to reversing flow and quantified the bristling process for individual scales using high-speed DPIV and a simple flow setup to generate velocities comparable to cruising conditions for a shark. Specific moments when scales were actuated by reversing flow were studied. It was documented that scale bristling was initiated when the predominant flow direction locally changed to a flow reversal, with a median velocity magnitude near the scale of 0.33 m·s^−1^. The average maximum angle of the scale bristling observed was 42 ± 7° once the reversing flow reached its maximum, and the entire bristling process took 1.7 ± 0.8 m·s^−1^ [7].

This study focuses on determining the spanwise spacing and width of the low-speed streaks in the presence of an APG with and without separation in an experimental water tunnel study. The findings were correlated with the width of the scales from the flank region. Moreover, the experiments were also carried out over shark skin samples. The ultimate goal was to fully understand the dimensional sizing requirements to design and manufacture shark-skin-inspired microgeometries that mimic the movable shark scales capable of controlling flow separation using this passive flow-actuated control mechanism [3].

## 2. Experimental Setup

The experiments were carried out in a water tunnel with a test section measuring 38 cm × 76 cm × 275 cm (W × H × L) with a maximum flow speed of 0.7 m·s^−1^. A vertical smooth flat plate (SFP) measuring 45.72 cm × 256.84 cm (H × L) was placed in the test section (Figure 4). The SFP is made up of four black plexiglass panels, each 45.7 cm long by 61 cm wide, such that one panel can be easily replaced by a different one on which the shark skin specimen was mounted. The SFP includes an adjustable flap at the trailing edge (set at φ = 13.3° as shown in Figure 4) to prevent flow separation at the elliptical leading edge (LE).

The formation of an adverse pressure gradient (APG) at a strategic location on the flat plate was accomplished by using a rotating cylinder mounted parallel to the plate as shown in Figure 4. The cylinder is 65 cm in length and 5.1 cm in diameter, spanning the entire width of the flat plate to minimize secondary flow resulting from its rotation. The origin of the streamwise direction (x-axis) of the coordinate system for data presentation is set at the center of the cylinder, while the origin of the wall-normal direction (y-axis) is defined at the surface of the plate (Figure 4). The cylinder was positioned at 1.2 D from the flat plate and a distance Lc = 220.8δ* from the LE (where δ* is the boundary layer displacement thickness at the beginning of the measurement window for the smooth flat plate without the cylinder and equal to 6.6 mm downstream from the leading edge, such that if a separation point forms, it is downstream of the measurement region).

The use of a rotating cylinder is a reliable method to create an APG that does not require additional mechanisms, such as suction or blowing apparatuses, to induce flow separation along a flat plate. Moreover, the separation location induced on the plate downstream of the cylinder depends directly on the cylinder’s RPM (Table 1) and/or the distance between the cylinder and the flat plate [5,32,33]. All vortex shedding from the cylinder was suppressed for all Re_x_ tested by maintaining a ratio of rotational speed to free stream velocity ($\alpha =\frac{f\Omega D}{U_{0}}$, where α is the velocity ratio, f is the frequency, D is the diameter, and U_0_ is the free stream velocity) of greater than 2.0 [34]. The rotation also causes the cylinder’s wake to move away from the plate, preventing any interaction between the wake and the boundary layer of the flat plate [5].

This study compares the flow over a smooth flat plate (SFP) to that of a shark skin specimen from a region prone to separation due to the shark’s streamlined body (flank region), which has the most flexible scales (bristling angle of 50°) [31]. Four patches of shark skin specimen from the flank were affixed to the test panel. Each patch has dimensions of 14 cm in length and 54 cm in height (88.4% of the total height of the panel). A smooth transition from the SFP to the test panel containing the shark skin specimen was ensured by eliminating any gaps using waterproof tape (Figure 5). Special care was taken while handling the shark skin test panel to avoid any scale damage by contact. Moreover, when not in use, the specimen was kept frozen and allowed to thaw before testing again. Turbulent flow was induced in the boundary layer by placing a round trip wire at L_t_ = 63δ* from the LE (Figure 4). The Reynolds number based on the diameter of the trip $\left({Re}_{D}=\frac{U_{0}D_{wire}}{\nu }\right)$ was greater than 826 to induce transition right after the trip [35,36] for the lowest freestream velocity tested.

The flow field was measured using digital particle image velocimetry (DPIV). A parallel laser sheet to the flat plate (x–z plane) with a pulse repetition rate of 1.0 kHz was generated by an Nd-YLF laser (Quantronix Darwin 527-30-M laser; power = 30 Watts, pulse energy = 20 mJ). The flow was seeded with 14 μm of neutrally buoyant silver-coated hollow glass spheres and images were captured by a high-speed CCD camera (Basler A504K with Nikon AF Micro Nikkor 105 mm lens, 1280 × 512 resolution, running at 1000 fps), to image the particles illuminated within the laser sheet for a 3.5 cm × 8 cm measurement region.

The camera was positioned at x = 7δ* to capture the LSS formation in the presence of an APG. As previously discussed, it was expected that the average spanwise spacing of the LSS would increase, and if separation occurs downstream, the LSS would disappear within the separation region and reform downstream of the reattachment [30]. However, it was expected that the LSS would be maintained over the shark skin specimen since the scales can delay or eliminate separation.

Data acquisition consisted of taking 40,000 images (40 s of data acquisition) for three different Re_x_ = 4.95, 6.1, and 7.1 × 10^5^. The Re_x_ choice and experimental setup were the same as that of a previous study [10]. The laser sheet was positioned parallel to the plate at y^+^ = 16, 24, 33, and 41 (see Table 1). TSI Insight^4G^ DPIV software (version 11.2.1.0) was used to process the raw images, and a MATLAB script was used for post-processing and analyzing the data. Image calibration resulted in (79 pixels/cm). Once the background noise was eliminated using a background image filter, all images were processed with a Recursive Nyquist Grid, FFT Correlation Engine, and Gaussian Peak Engine with a large interrogation window of 40 × 40 pixels followed by a small interrogation window of 10 × 10 pixels. The small interrogation window was needed to capture the turbulent structures near the wall. A post-processing filter reduced the measurement error caused by localized insufficient seeding. Poor seeding, image quality, camera noise, and particle displacement gradient are the main factors associated with errors in DPIV [37,38,39]. Each image had approximately 23,000 vectors on average, of which 4% were invalid vectors, which is in good agreement with allowable DPIV error [37,38]. Nevertheless, the proximity to the wall can increase the DPIV error, as indicated by previous studies [39,40], and hence the additional processing with the smaller window size was carried out to increase measurement resolution. The seeding density was kept uniform during the experiment to minimize any errors. The range of the DPIV optimum seeding density is around 0.02 to 0.04 ppp [38]. In this experiment, the seeding density was 0.024 ppp for a 10 × 10 window near the wall according to TSI Insight 4G™ software. The total velocity field vector error was around 6% after post-processing.

## 3. Results and Discussion

In this experiment, flow measurements captured LSS formation upstream of the separation point for the flow over the SFP and over shark skin for Re_x_ ranging from 4.9 × 10^5^ to 7.1 × 10^5^. The turbulent boundary layer separation occurred at a distance of 11δ* from the cylinder center, as indicated by the backflow coefficient in Figure 6a for Re_x_ of 4.9 × 10^5^. The backflow coefficient is defined as the fraction of the total time that the flow is reversed at a given location [41,42]. Boundary layer separation is defined when the backflow coefficient exceeds 50% and the skin friction coefficient is zero. The focus of this study was on LSS formed upstream of the separation (5 < X/δ* < 9). Moreover, the separation region was eliminated when the shark skin was mounted to the plate, as shown by the backflow coefficient in Figure 6b.

Figure 7 illustrates the backflow coefficient upstream of separation for a plane parallel to the plate at a distance of Re_x_ = 4.95 × 10^5^. In Figure 7a, it can be observed that the backflow coefficient did not exceed 31% over the SFP. In contrast, the flow over the shark skin specimen exhibited a maximum backflow coefficient of less than 10%, as depicted in Figure 7b. As expected, the backflow coefficient was higher for the flow upstream of the separation point. The reduction in the backflow coefficient from 31% (over the SFP) to 6% (over shark skin) suggests that the shark scales significantly influence reversing flow formation and delay separation.

The experimental pressure gradients were derived from the DPIV data (Figure 8) using the pressure Poisson equation (Equation (1)) [43]. While this equation offers a means to estimate the pressure gradient, it is important to acknowledge the possibility of accumulating measurement errors stemming from the required temporal and spatial derivatives of streamwise velocity (u). The experimental pressure gradient may also be underpredicted due to DPIV measurement error [39,40]. On the other hand, an inviscid theoretical model can overpredict the pressure gradient due to not accounting for viscous effects [32]. The calculated experimental pressure gradient was 11.2%, 10.3%, and 12.52% lower than the theoretical model prediction for Re_x_ ≈ 4.95, 6.1, and 7.1 × 10^5^, respectively.
(1)$$\nabla p=-\rho \left\{-\left(\bar{u}.\nabla \right)u^{\prime }+\left(u.\nabla \right)u-\nu \nabla^{2}u\right\}$$

Direct numerical simulation (DNS) findings over the SFP reveal that the LSS average spanwise spacing ($\bar{\lambda^{+}}$) at the wall upstream of the separation point is greater than 100 [30]. In contrast, the LSS average spanwise spacing for a turbulent flow near the wall without separation is roughly 100 ± 20 [15], gradually increasing with distance from the wall [26]. The rise is caused by the elongation of the streamwise vortex legs of the hairpin vortices as they move away from the surface [19]. Since the shark scales from the flank region can control flow, the streaks are expected to have an average spanwise spacing ($\bar{\lambda^{+}}$) near the wall of approximately 100 for these cases, which is more consistent with the unseparated flow case.

**Figure 8 biomimetics-09-00378-f008:**
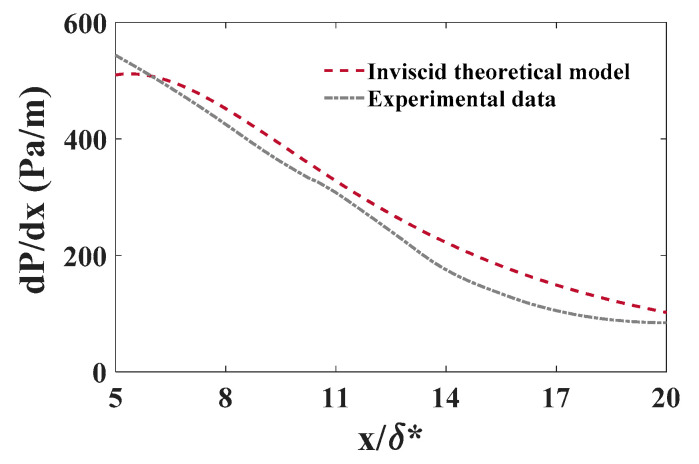
The experimental predicted and theoretical inviscid model of the pressure gradient induced on the boundary layer for Re_x_ ≈ 4.95 × 10^5^. δ*—boundary layer displacement thickness; x—streamwise distance from the center of the cylinder.

The locations of reversing flow in the LSS and surrounding vortices were determined using an in-house Root Mean Square (RMS) Intersect method. This innovative approach analyzes the normalized RMS of velocity fluctuations within a 2-D flow relative to local mean velocities. The interception of normalized RMS values along streamwise and normal directions represents the center of a fully developed vortex. Figure 9 represents the LSS at a distance y^+^ = 16 away from the wall. In Figure 10, the velocity vector field is overlaid onto a reversing flow within a LSS to illustrate the precise locations of these vortices. Moreover, Figure 11 depicts the RMS Intersect method applied to the flow field depicted in Figure 10. This method offers a robust means of identifying vortices and reversed streak boundaries, which is crucial for statistical analysis and further understanding of the fundamental fluid dynamics.

The probability density function (PDF) of the LSS spanwise spacing for Re_x_ = 4.95 × 10^5^ over the SFP at four different distances from the wall is shown in Figure 12. These locations (y^+^) vary from the buffer layer to the beginning of the logarithmic layer. Each histogram represents approximately 115 streak samples acquired in a period of 40 s. In the lognormal probability density function in Figure 12, the average LSS spanwise spacing ($\bar{\lambda^{+}}$) grows linearly as the distance from the wall (y^+^) increases (18 $\bar{\lambda^{+}}$ units for every 8-unit rise in y^+^ from the buffer layer until it reaches the beginning of the logarithmic layer). The average spanwise spacing ($\bar{\lambda^{+}}$) increases from 170 (at y^+^ = 16) to 223 (at y^+^ = 41) (Figure 12). The average spanwise spacing for a turbulent flow over an SFP with a zero-pressure gradient (no separation) is expected to be 116 at y^+^ = 16 and 144 at y^+^ = 33 from the wall [22]. In the current experiment, the average spanwise spacing exceeds those of a flat plate at zero pressure gradient due to the APG [26]. The rise in the spanwise spacing standard deviation as the distance from the wall increases (y^+^) (see Figure 12) implies a broadening of the streak distribution [20,22]. Figure 12 shows that the current experimental study agrees with DNS results [30], which indicate that the LSS spanwise spacing increases in the presence of an APG.

In Figure 13, PDF histograms of the low-speed spacing are compared between the SFP and shark skin specimens at the plane closest to the wall (y^+^ = 16) for variation in Re_x_. The first observation is that the average spanwise spacing and standard deviation increased as the Re_x_ increased, and this is likely due to the increase in APG (increased cylinder rotation rate) that was required to induce separation as flow velocity increased. However, the average spanwise spacing and its deviation are clearly much greater over the SFP upstream of the separation than over the shark skin specimen. The difference in the LSS average spanwise spacing indicates that the shark scales are able to control the flow separation by controlling the flow in the LSS [21]. The bristling scales near the wall disrupt reversing flow, preventing it from gaining strength and size, such that the forward momentum of the oncoming flow is maintained, thus keeping the flow attached.

Likewise, the width of the low-speed streaks $\left(s^{+}=\frac{su_{\tau }}{\nu }\right)$, where s is the width of the LSS, becomes thinner when the shark skin specimen is mounted in the test section (Figure 14b) for all Re_x_ tested. The average LSS width is 1.5 times greater than the flow over the SFP upstream of separation (Figure 14a). This result is consistent with the DNS results, which demonstrate that the streaks become wider and the spanwise spacing becomes larger for a separated flow versus an attached turbulent boundary layer [30].

The Root Mean Square Intersect Method revealed that most LSS are surrounded by two or three vortices, which agrees with the hairpin vortex signature model for a typical hairpin packet [25]. Most of the vortices’ centers are located between 40% and 80% of the measured streak length for all Re_x_ tested in this experiment (Figure 15a). Furthermore, the mean streamwise spacing between the vortices’ centers along the LSS varies from 0.54δ* to 1.08δ* (Figure 15b). This result is in accordance with the estimated streamwise spacing between hairpin heads from a previous experiment to validate the hairpin packet theory [44,45].

According to the hairpin packet theory, the streamwise and spanwise vortices help to push the flow upstream, generating the patch of reversed flow [25]. The streamwise reversing flow magnitude within the streaks gradually increases from the edge toward the center of the streaks. We hypothesize that the width of the highest negative streamwise velocity (reversed flow) patch is responsible for bristling the shark scales that impede further flow reversal to control the flow.

Characteristic instantaneous flow measurements of streaks upstream of separation are shown in the top row of Figure 16a–c and over the shark skin in the bottom row of Figure 16d–f. The spanwise spacing increases as the Reynolds number increases from 4.95 × 10^5^ (first column Figure 16) to 7.1 × 10^5^ (last column Figure 16). The number of streaks representing the reversed flow (in blue Figure 16) over the shark skin specimen is higher than over the SFP upstream of the separation point. The reduction in the number of streaks upstream of the separation is well aligned with existing literature [30]. As reported, these streaks gradually decrease in number as they approach separation and eventually vanish within the separation region. Figure 17, Figure 18 and Figure 19 compare one LSS of each subplot in Figure 16 for the width of the highest negative velocity within the reversed streak between the SFP and shark skin specimens for each Re_x_ tested (each column in Figure 16). The velocity vector is added to each subplot to show the vortices along the highest shear layer (the line between positive and negative velocities). In addition, the normalized streamwise velocity contour scale range is adjusted from (−0.2 to 0.4) to (−0.30 to 0) by 0.02 velocity increments to better represent the difference in velocities within the reversed streaks. The white color represents the positive streamwise velocity and the black horizontal grid lines are 8.26 viscous units ($\lambda^{+}$) apart for the lowest Re_x_ and 10.8 for the highest Re_x_ tested.

In this experimental setup, the reversed velocity should be around 20–30% of the free stream velocity [30] to induce bristling of the scales. The width of crown shark scales from the flank region (170 µm) is approximately 1.1 to 1.2 viscous length units for the range of three Re_x_ tested in the current study. This is due to the fact that the flow speeds being tested are much lower than those that would occur over a real shark. For the lowest Re_x_ tested, the top streak for the flow over the SFP upstream of the separation (Figure 16a) and the top streak for the flow over the shark skin (Figure 16d) are analyzed more closely and shown in Figure 17a and Figure 17b, respectively. The highest negative streamwise velocity is higher than 30% for the flow upstream of the separation over the SFP and around 26% of U_0_ for the flow over the shark skin specimen. In Figure 17a, the highest negative velocity patch has a width of 6.15 viscous length scales and is 5.7 times greater than the viscous length scale for the width of the shark scale crown (width of the streaks compared to the width of the shark scale crown in viscous length scale units). This viscous length scale ratio decreases from 5.7 to 5.5 for the streaks over the shark skin specimen (Figure 17b). The highest negative velocity observed within the streaks over the shark specimen is consistent with the minimum predicted value that is required to bristle the scales.

In Figure 18, the highest negative streamwise velocity is approximately 28% of U_0_ for the flow over the shark scales and over 30% for the flow over the SFP for the Re_x_ of 6.1 × 10^5^. The vortices are observed along the streaks in the shear layer with the highest shear stress. The streak viscous length scale over the SFP upstream of separation is 8.9 and 6.8 over the shark skin specimen and the viscous length scale ratios (streak width/shark scale crown width) are 7.7 for the SFP experiment and 5.9 when the shark was mounted in the separation region. For the last case tested (Re_x_ of 7.1 × 10^5^), the viscous length scale is 10.8 for the flow over the SFP with separation and 7.2 over the shark skin when mounted to the smooth flat plate (Figure 19). The width ratios are 8.9 and 5.9 for the flow over the SFP and over the shark scales, respectively. Again, the reversed streaks for the flow over the shark scales are thinner than those upstream of the separation for each Re_x_ tested. Despite variation in the Re_x_, all shark skin cases exhibit a consistent ratio of around 5.9 for the streak width to shark scale crown width (Table 2). Moreover, the experiment reveals that the highest velocity within the reversed patch falls within the flow velocity range required to bristle the scales, and velocities higher than this, as occurred in the SFP case, were not detected. This observation validates the hypothesis that reversing flow in the LSS causes shark scales to bristle, thereby controlling flow on the verge of separation. In addition, the ratio represents the number of scales that could be bristled by the highest negative streamwise velocity since the rows of shark scales are not perfectly aligned, as shown by the electronic microscope image in Figure 20a.

Furthermore, for reader comprehension the results from a previous study [7] conducted over a shark skin specimen to visualize scale bristling are shown here so the reader can view scales at rest (Figure 20b) and a few scales that have been bristled by a reversed flow (Figure 20c). In the current study, the number of bristled scales in the same spanwise row would be at least 6 based on the width of the highest negative velocity.

## 4. Conclusions

Previous experiments have proven that real shark skin in water tunnel experiments can control flow separation. Even though there appears to be some Re_x_ independence to the function of the separation control mechanism, as long as the boundary layer thickness is comparable and sufficient reversing flow velocities are obtained to achieve scale bristling, separation control due to movable scales impeding reversing flow close to the surface is the fundamental mechanism and is still documented in water tunnel studies over shark skin specimens. It is hypothesized that the separation control function of sharks’ scales may be more effective at typical shark swimming speeds.

It should be mentioned that for flow over a real shark swimming at higher speeds, the viscous length scale will be significantly reduced. This means that instead of 6 or more scales being bristled within an LSS, it will likely only be one at burst swimming speeds. However, due to limitations of the flow speed for the water tunnel, the shark skin had to be tested at lower flow speeds. It is hypothesized that the shark skin functions more effectively at the Re_x_ occurring over a real shark. However, the dimensional similarity that was preserved in these experiments is the height to which the scales bristle into the flow because the boundary layer thickness was similar as to what would occur over a real shark. This would indicate that a biomimetic surface would need to be sized according to the highest Re_x_ application but could still potentially control flow separation at lower Re_x_ as well. The fundamental control mechanism is scale bristling that impedes reversing flow, whether that is accomplished by a single scale or multiple scales in a spanwise row.

To reiterate, it is believed that the reversing flow occurring in the LSS for a turbulent boundary layer under the influence of an APG is the mechanism that initiates scale bristling. This study analyzed the LSS formation upstream of the separation point for a flow over an SFP and found agreement with DNS results in literature. Furthermore, the LSS over the SFP were compared to those forming over a shark skin specimen to document the effect that the separation control mechanism of flexible scales has on the fundamental structure of the flow.

This experimental study agrees with the results of a DNS study [30] for the flow structure in a turbulent boundary layer under the influence of an APG. The LSS near the wall and upstream of the separation point have an average spanwise spacing 1.7 times greater than the flow with a zero-pressure gradient (100 ± 20), as reported in literature [46]. The LSS have at least two vortices along the high shear stress region, and the distance between the vortices’ centers validates the hairpin model. Further away from the wall, the spanwise spacing also increases. However, when the shark skin is mounted in the test section (where flow separation occurs over the SFP), the average spanwise spacing near the wall decreases to a value similar to a flow with a zero-pressure gradient. This decrease validates that the scales are not functioning as a traditional rough surface as otherwise, the average spanwise spacing would increase. The decrease is attributed to shark scales controlling the flow by reducing or eliminating flow separation. Even though the width of the streaks decreases over the shark skin specimen, as well as the highest negative velocity within the streaks over the shark scales, reversed velocities must still occur of sufficient magnitude to bristle the scales. But it should also be clear that at the same x location, the shark skin will have lower magnitudes of reversing flow when compared to the SFP case, which is on the verge of separation.

Finally, the ratio of LSS width (for the highest negative streamwise velocity) to shark crown width ratio was found to be around 6 for all shark skin cases. This value indicates the number of scales that are bristled simultaneously by the reversed flow to control the flow separation and the magnitude of reversing flow required to bristle the scales (measured here as the maximum reversing flow detected within LSS over the shark skin). This result aligns well with the flow-induced bristling of scales observed on real shark skin found in the literature [7].

## Figures and Tables

**Figure 1 biomimetics-09-00378-f001:**
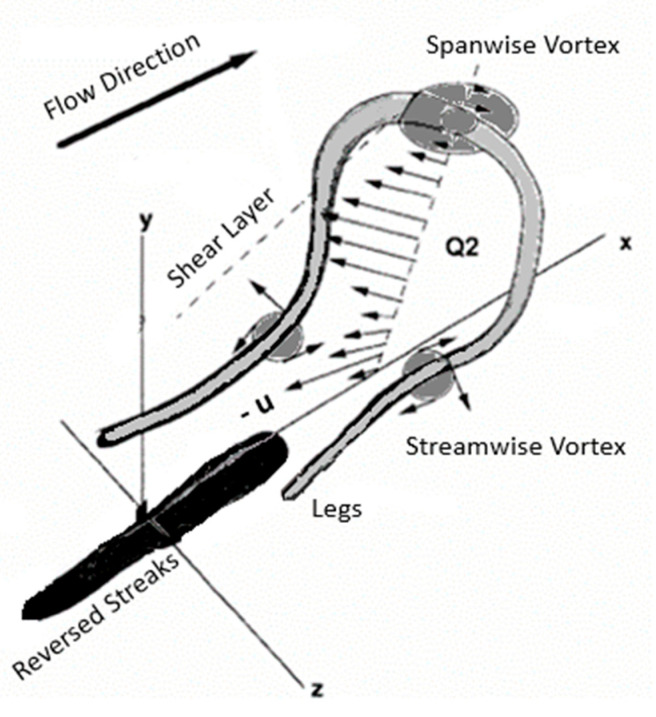
Signature of the hairpin vortex attached to the wall in the streamwise–wall-normal plane.

**Figure 2 biomimetics-09-00378-f002:**
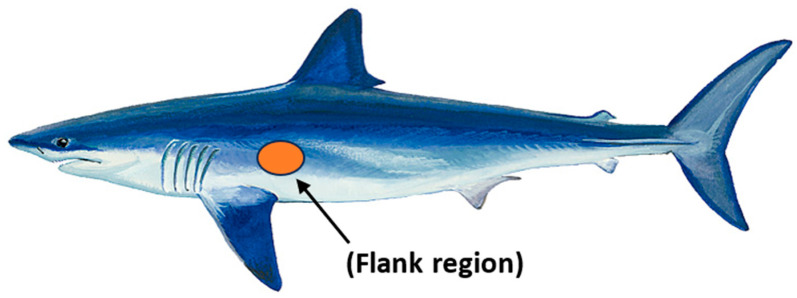
Shortfin mako, *Isurus oxyrinchus,* and the flank region located behind the gills that has the scales with the highest flexibility (>50°).

**Figure 3 biomimetics-09-00378-f003:**
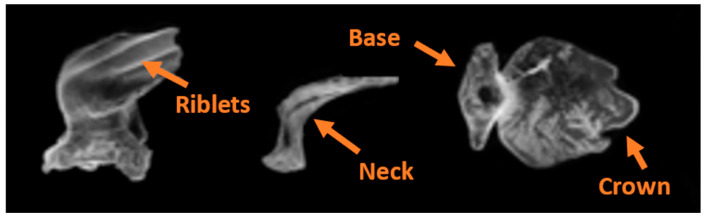
Microscopic electronic isometric, lateral, and ventral images of placoid scales from the flank area of a male shortfin mako measuring 158 cm in total length. Modified from Lang et al. (2011) [2].

**Figure 4 biomimetics-09-00378-f004:**
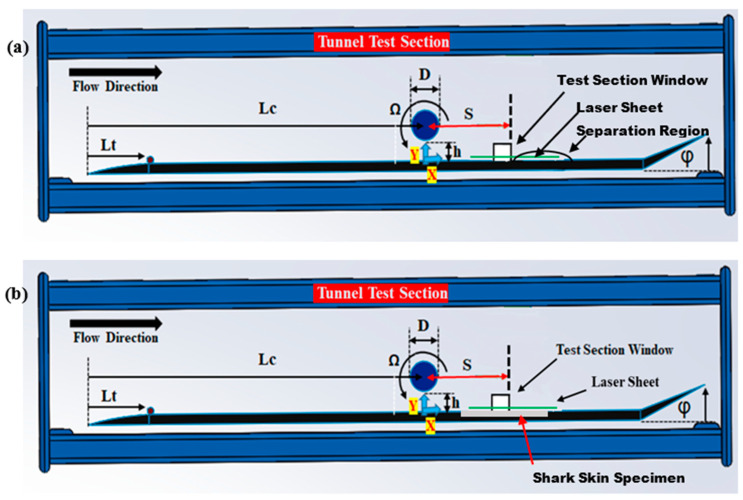
Top view of the tunnel test section. (**a**) Smooth flat plate experiment; the separation bubble dimensions are 3δ* in height and 26δ* in length. (**b**) Shark skin experiment. L_T_—distance from the trip to the LE; L_C_—distance from the center of the cylinder to the LE; h—the gap between cylinder and plate; S—distance from the center of the cylinder to the shark skin test section; D—diameter of the cylinder; φ—trailing edge angle; Ω—rotational speed of the cylinder, and δ*—boundary layer displacement thickness for the Re_x_ ≈ 4.95 × 10^5^. The origin is represented by the blue arrows.

**Figure 5 biomimetics-09-00378-f005:**
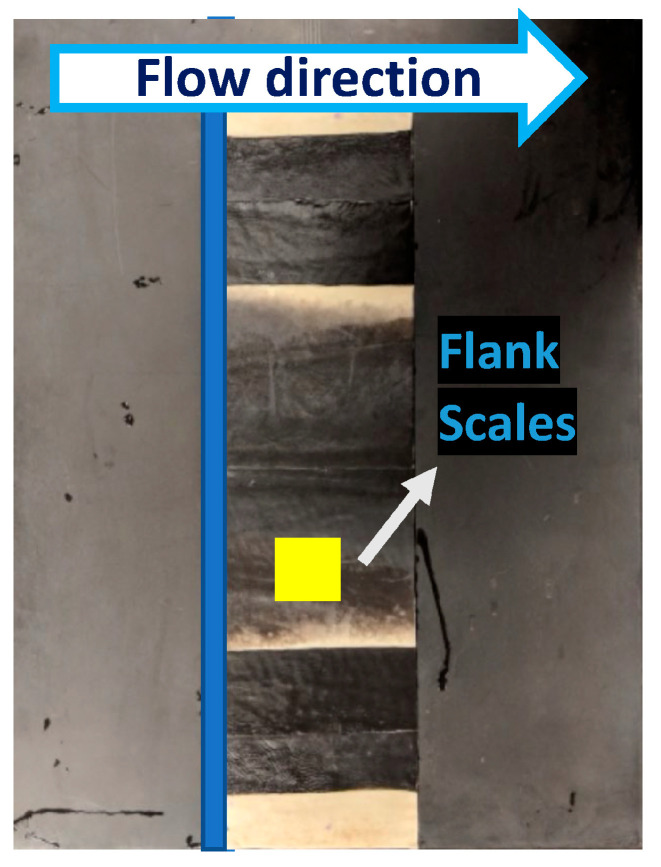
The experimental setup in the water tunnel. Four patches of shark skin specimen from the region behind the gills, which include regions (flank region) mounted in the test section. The third patch from the top was used to acquire the data. Modified from Santos et al. (2021) [10].

**Figure 6 biomimetics-09-00378-f006:**
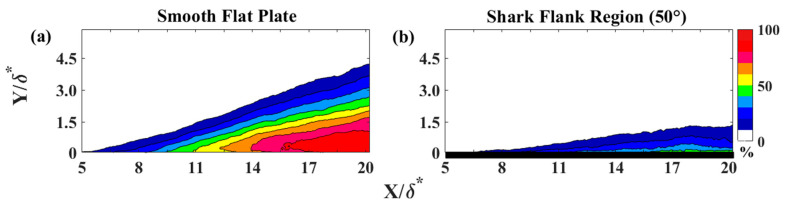
Backflow coefficient contour for Re_x_ ≈ 4.9 × 10^5^. (**a**) The dividing line between green and yellow contours represents a backflow coefficient of 50%, which is the minimum required threshold for separation. (**b**) The black bar at the bottom of column b represents the shark skin test section. x—streamwise distance from the center of the cylinder; y—perpendicular distance from the flat plate; and δ*—boundary layer displacement thickness for the Re_x_ ≈ 4.9 × 10^5^.

**Figure 7 biomimetics-09-00378-f007:**
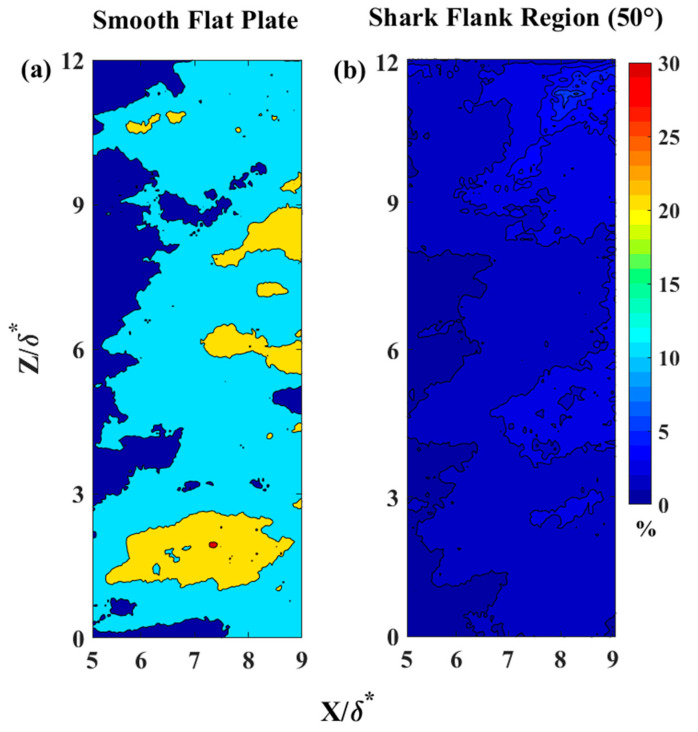
Backflow coefficient contour upstream of the separation at a distance of y^+^ = 16 away from the surface for Re_x_ ≈ 4.9 × 10^5^. (**a**) Flow over a smooth flat plate with a backflow scale from 0 to 30%; (**b**) flow over the shark skin. x—streamwise distance from the center of the cylinder; z—spanwise distance perpendicular to the x-y plane; and δ*—boundary layer displacement thickness for the Re_x_ ≈ 4.9 × 10^5^.

**Figure 9 biomimetics-09-00378-f009:**
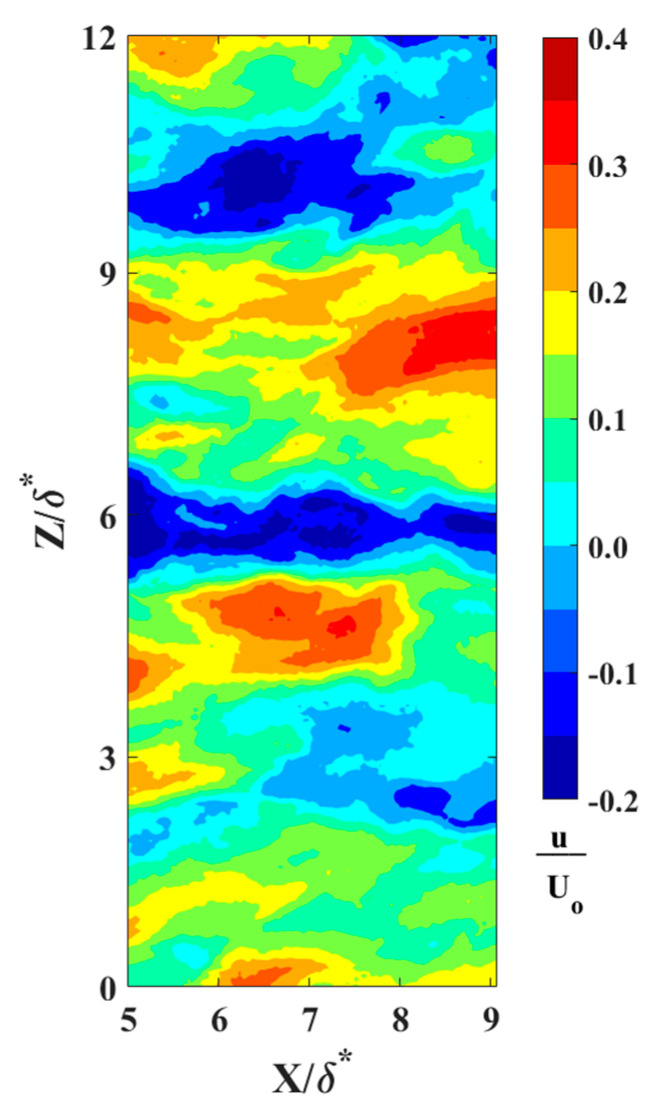
Low-speed streaks (streamwise reversed flow only) at a distance of y^+^ = 16 above the wall (Re_x_ = 6.1 × 10^5^). The flow field x-y plane is perpendicular to the wall.

**Figure 10 biomimetics-09-00378-f010:**
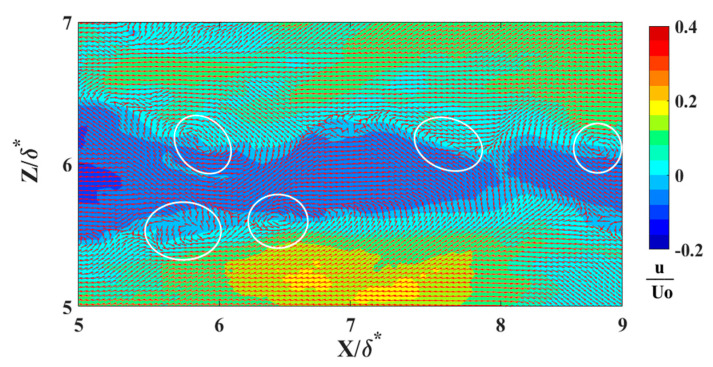
Low-speed streaks (streamwise reversed flow only) at a distance of y^+^ = 16 above the wall (Re_x_ = 6.1 × 10^5^). The flow field x-y plane is perpendicular to the wall. White circles represent the locations of the vortices along the streaks.

**Figure 11 biomimetics-09-00378-f011:**
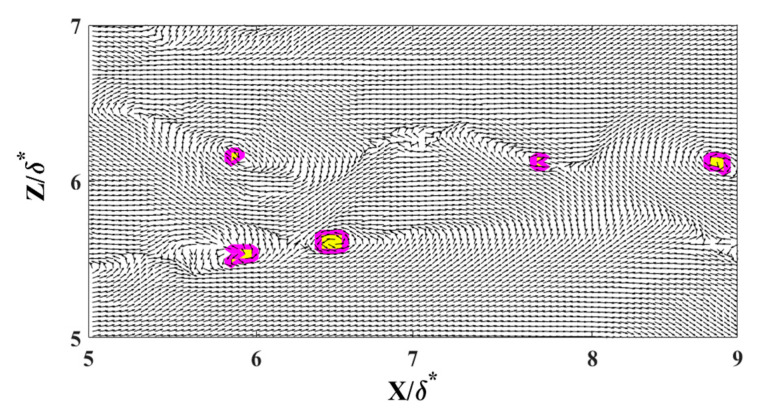
The RMS Intersect method is applied to the flow field represented in Figure 9. The magenta–yellow colors represent the location of the center of the vortices. The velocity vector field helps to verify the accuracy of the method.

**Figure 12 biomimetics-09-00378-f012:**
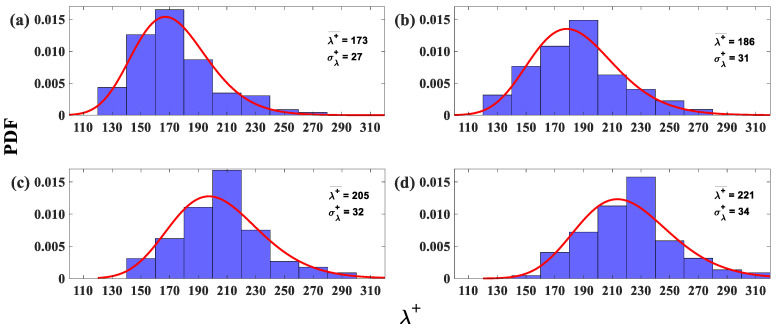
Probability density function (PDF) histograms and lognormal probability function (red curve) of the low-speed streak spanwise spacing (reversed flow only) for the flow upstream of the separation region at Re_x_ = 4.95 × 10^5^. (**a**) y^+^ = 16; (**b**) y^+^ = 24; (**c**) y^+^ = 33; and (**d**) y^+^ = 41. $\bar{\lambda^{+}}$—spanwise spacing average, $\sigma^{+}$—standard deviation average; and λ—spanwise spacing between low-speed streaks.

**Figure 13 biomimetics-09-00378-f013:**
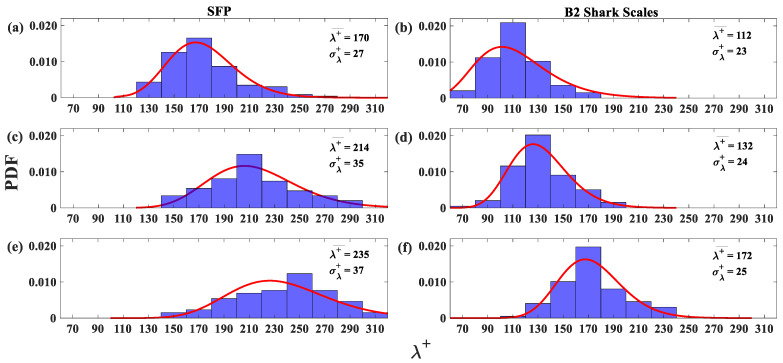
Probability density function (PDF) histograms and lognormal probability function (red curve) of the low-speed spacing (reversed flow only). The first column represents the streaks over the SFP upstream of the separation and the second column represents the streaks over the shark skin specimen. (**a**,**b**) Re_x_ = 4.95 × 10^5^; (**c**,**d**) Re_x_ = 6.1 × 10^5^; and (**e**,**f**) Re_x_ = 7.1 × 10^5^. $\bar{\lambda^{+}}$—spanwise spacing average; $\sigma^{+}$—standard deviation average.

**Figure 14 biomimetics-09-00378-f014:**
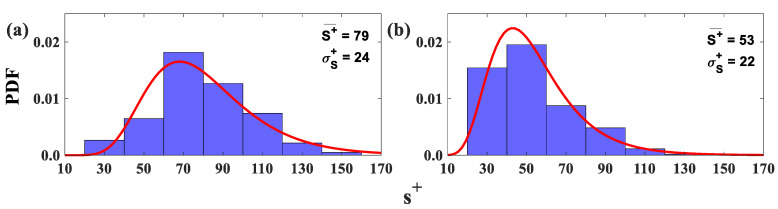
Probability density function (PDF) histograms and lognormal probability density function (red curve) of the width of the low-speed regions (reversed flow only) at a distance of y^+^ = 16 from the wall for all Re_x_ tested. (**a**) Flow over the SFP upstream of the separation. (**b**) Flow over the shark skin specimen.

**Figure 15 biomimetics-09-00378-f015:**
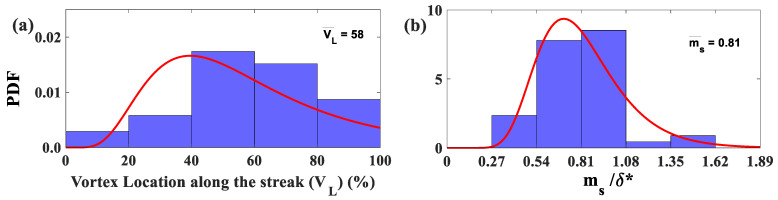
(**a**) Probability density function (PDF) histograms and lognormal probability density function (red curve) of the vortex’s center location within the low-speed streaks (reversed flow only). (**b**) Streamwise spacing of the adjacent vortices’ centers along the low-speed streaks. All Re_x_ were tested at a distance of y^+^ = 16 above the wall.

**Figure 16 biomimetics-09-00378-f016:**
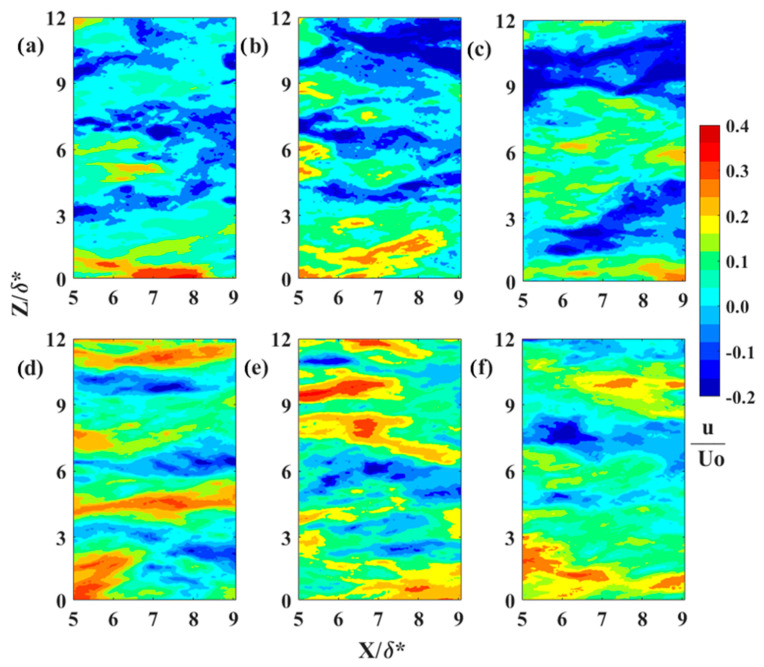
Low-speed streaks for (**a**,**d**) U_0_ = 0.33 m·s^−1^ at Re_x_ 4.9 × 10^5^, (**b**,**e**) U_0_ = 0.40 m·s^−1^ at Re_x_ 6.1 × 10^5^, and (**c**,**f**) U_0_ = 0.47 m·s^−1^ at Re_x_ 7.1 × 10^5^. The first row represents the flow over the SFP upstream of the separation and the second row represents the flow over the shark skin specimen. The flow moves from the left to right and the blue color represents the reversed flow. U_0_—the freestream velocity; X—streamwise distance from the center of the cylinder; Y—perpendicular distance from the flat plate; and δ*—boundary layer displacement thickness for the Re_x_ ≈ 4.9 × 10^5^.

**Figure 17 biomimetics-09-00378-f017:**
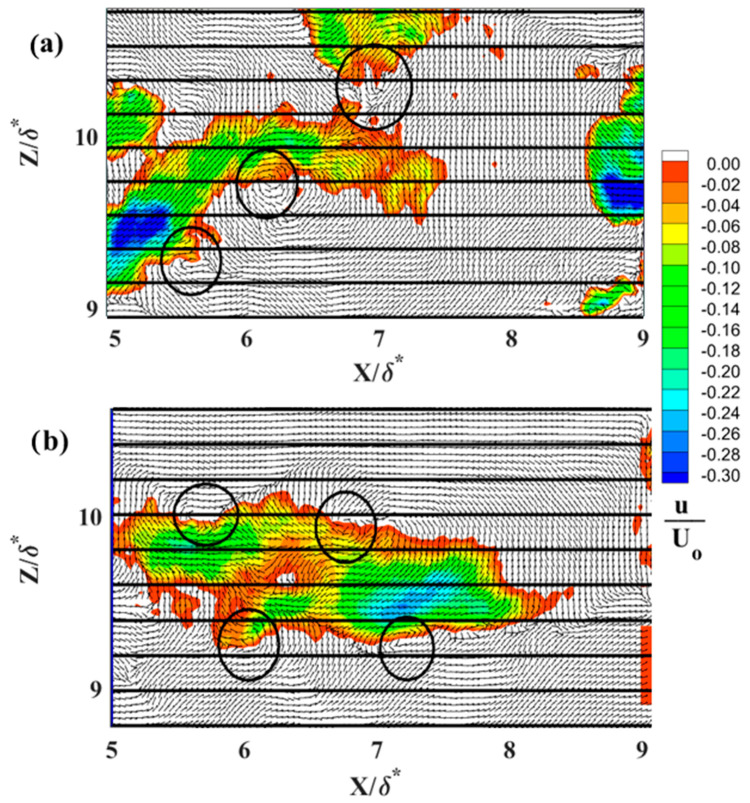
(**a**) LSS over a smooth flat plate upstream of the separation (top streak is from Figure 16a); (**b**) LSS over the shark skin specimens (top streak is from Figure 16d). The black circles represent the location of the vortices; the black horizontal lines are 8.26 viscous units apart at Re_x_ of 4.95 × 10^5^.

**Figure 18 biomimetics-09-00378-f018:**
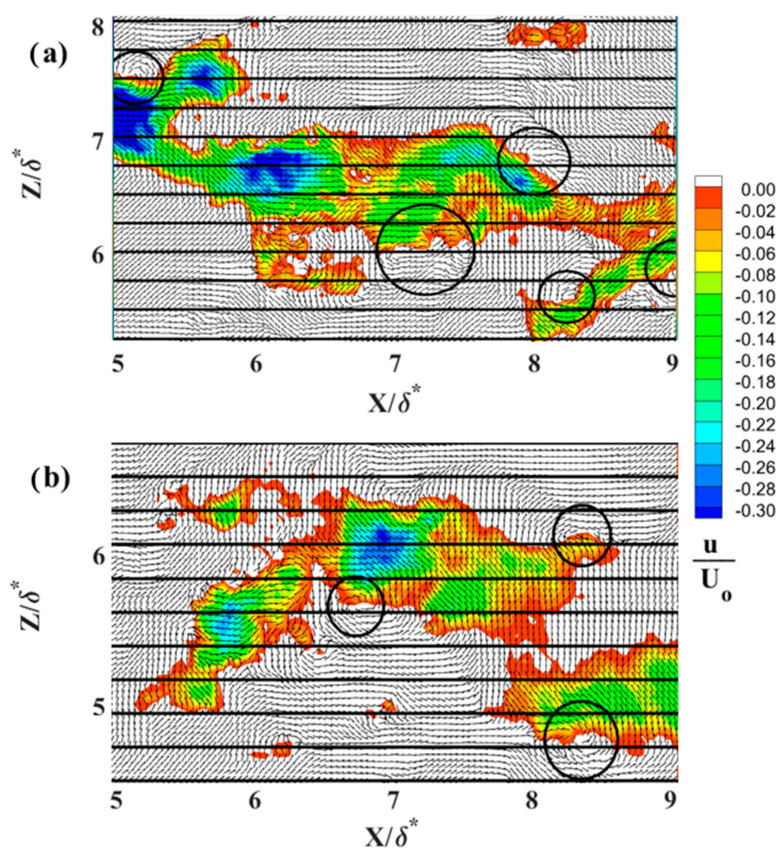
(**a**) LSS over a smooth flat plate upstream of the separation (middle streak is from Figure 16b); (**b**) LSS over the shark skin specimens (middle streak is from Figure 16e). The black circles represent the location of the vortices; the black horizontal lines are 8.92 viscous units apart at Re_x_ of 6.1 × 10^5^.

**Figure 19 biomimetics-09-00378-f019:**
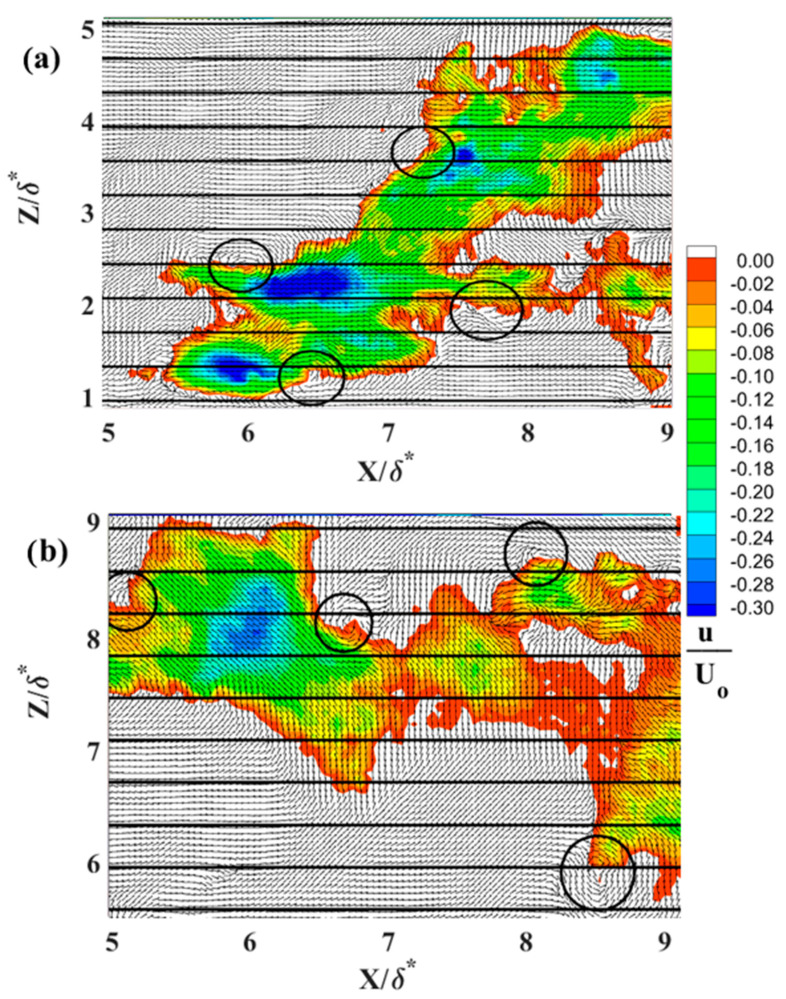
(**a**) LSS over a smooth flat plate upstream of the separation (bottom streak is from Figure 16c); (**b**) LSS over the shark skin specimens (middle streak is from Figure 16f). The black circles represent the location of the vortices; the black horizontal lines are 10.8 viscous units apart at Re_x_ of 7.1 × 10^5^.

**Figure 20 biomimetics-09-00378-f020:**
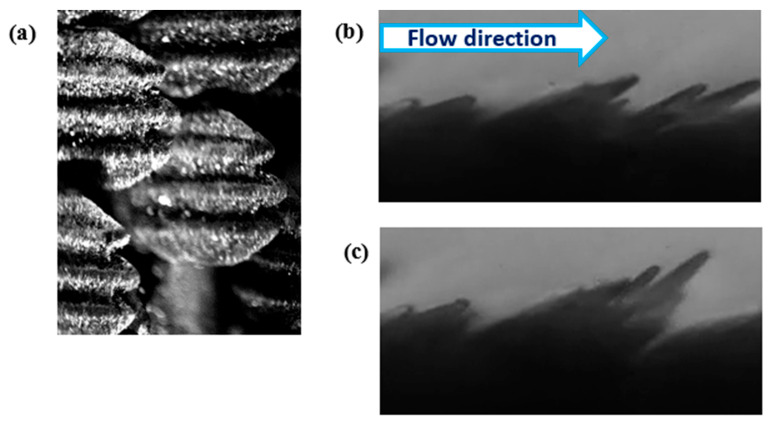
(**a**) Top view of mako shark scales from the flank region; (**b**) shark scales at rest; and (**c**) shark scales bristled due to a reversed flow generated by forward jet.

**Table 1 biomimetics-09-00378-t001:** Smooth flat plate boundary layer flow parameters, the cylinder rotation, and the wall units.

Re_x_	Ω (RPM)	U_0_ (m·s^−1^)	u_τ_ (mm·s^−1^)	y^+^
4.95 × 10^5^	420	0.33	8.26 (SFP), 6.39 (Shark)	16, 24, 33, 41
6.1 × 10^5^	520	0.40	8.92 (SFP), 6.81 (Shark)	
7.1 × 10^5^	630	0.47	10.8 (SFP), 7.20 (Shark)	

**Table 2 biomimetics-09-00378-t002:** Comparison of the width of the highest reversed velocity within the streaks to the shark scale crown width in viscous units for all Re_x_.

	Viscous Length			Ratio Streak Width/Shark Crown Width	
Re_x_	Crown Width	Reversed Streak Width over Smooth Flat Plate	Reversed Streak Width over Shark Scales	Streak over the SFP	Streak over the Shark Scale
4.95 × 10^5^	1.08	6.2	6.0	5.7	5.5
6.1 × 10^5^	1.15	8.9	6.8	7.7	5.9
7.1 × 10^5^	1.21	10.8	7.2	8.9	5.9

## Data Availability

The original contributions presented in the study are included in the article, further inquiries can be directed to the corresponding author/s.

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
