# Peer review of "Understanding Low-Speed Streaks and Their Function and Control through Movable Shark Scales Acting as a Passive Separation Control Mechanism"

_biomimetics, 2024, doi:10.3390/biomimetics9070378_

Round 1

Reviewer 1 Report

Comments and Suggestions for Authors

This manuscript presents experimental measurements of low-speed streaks in turbulent boundary layers over shark skin. They find that the shark scales help delay separation by resisting reversal of low-speed streaks. The results are detailed and cases without sharks skin are compared with DNS simulations to help lend confidence to the experimental techniques and setup. The findings will interest many readers.

My only comment is that in section 2 (possibly in figure 4) a schematic of where the separation bubble forms and its size with respect to the location of the cylinder would be helpful. Also, it would helpful to indicate the size and location of the patch with shark scales with respect to where the separation bubble forms in the case without scales. Perhaps the figure could be duplicated with the top showing the separation and the bottom showing the location of the shark skin and the lack of separation?

Author Response

I really appreciate your advice and I have updated Figure 4 to include both the separation bubble and shark skin specimen mounted in the test section, with the dimensions of the separation bubble detailed in the figure description. Additionally, I have revised Figure 1 to ensure compliance with licensing permissions.

Reviewer 2 Report

Comments and Suggestions for Authors

The influence of the passive bristling of a shark's movable scales on the flow separation is investigated experimentally. 

Interesting results were obtained confirming the possibility of controlling separation in this way.

Minor issue

The position of the shark's skin in Fig. 4 is unclear

Author Response

Thank you for your advice. I have updated Figure 4 to include both the separation bubble and the shark skin specimen mounted in the test section, with the dimensions of the separation bubble detailed in the figure description. Additionally, I have revised Figure 1 to ensure compliance with licensing permissions.